ecology

stable isotopes, Arctic cetaceans, carbon, nitrogen

**Authors for correspondence:**
Marie Louis
e-mail: marie.louis@sund.ku.dk
Eline D. Lorenzen
e-mail: elinelorenzen@sund.ku.dk

# Population-specific sex and size variation in long-term foraging ecology of belugas and narwhals

Marie Louis[1], Mikkel Skovrind[1], Eva Garde[2],
Mads Peter Heide-Jørgensen[2], Paul Szpak[3]
and Eline D. Lorenzen[1]

[1]Globe Institute, University of Copenhagen, Denmark
[2]Greenland Institute of Natural Resources, Copenhagen, Denmark
[3]Trent University, Peterborough, Ontario, Canada

ML, 0000-0002-4611-5503; MS, 0000-0002-5430-5884;
EG, 0000-0003-3016-5722; MPH-J, 0000-0003-4846-7622;
PS, 0000-0002-1364-6834; EDL, 0000-0002-6353-2819

Intraspecific variation in resource use by individuals of different age, sex or size may reflect differing energetic requirements and physiological constraints. Males and females often show differences in diet owing to sexual size dimorphism, different life histories and/or habitat use. Here, we investigate how sex and size influence the long-term foraging ecology of belugas and narwhals in Greenland, using stable isotopes of carbon and nitrogen from bone collagen. We show that males have a higher trophic level and a larger ecological niche than females in West Greenland belugas and in East Greenland narwhals. In addition, for these two populations, we find that $\delta^{15}$N increases with size, particularly in males. We hypothesize that sexual size dimorphism together with strong maternal investment drive these differences. By contrast, we find no differences in foraging ecology between sexes in West Greenland narwhals and observe no influence of size on trophic level. This may reflect the influence of interspecific competition in West Greenland, where the distributions of belugas and narwhals overlap, and/or geographical resource partitioning among different summer aggregations of narwhals. Our results suggest that sex and size variations in diet are population dependent, and probably the result of varying ecological interactions.

# 1. Introduction

The foraging ecology of a population may be driven by the abundance and distribution of resources, interspecific competition,

**Figure 1.** (*a*) Sample localities (a: Qaanaaq, b: Melville Bay, c: Kullorsuaq, d: Nuussuaq, e: Uummannaq, f: Qeqertarsuaq, *g*: Scoresbysund) with sample size and distribution of belugas and narwhals [29]. The WG narwhal population comprises samples from four localities. Coastline data are from [30]. (*b*) Bone $\delta^{13}$C and $\delta^{15}$N for male and female WG belugas ($n_M =$ 14, $n_F = 9$), WG narwhals ($n_M = 19$, $n_F = 20$) and EG narwhals ($n_M = 22$, $n_F = 16$). Solid circles indicate Bayesian standard ellipse areas (SEA$_B$). Mean (square) and s.d. (error bars) are indicated.

and age and sex structure within the population [1–4]. Animals of different age, sex or size may show variations in resource use such as preferred prey or foraging habitat, owing to differing energetic requirements and physiological capabilities [5–7]. This may act to reduce intraspecific competition [2]. In mammals, strong sexual size dimorphism and/or different life histories, including maternal care, between males and females, can often lead to dietary differences [7]. Females raise their young for several weeks to years and adjust their foraging behaviour to ensure offspring survival [8].

In the marine environment, differences in foraging ecology between sexes are particularly strong in pinnipeds [8–11], where males are often heavier and larger than females [12]. It is thought that larger individuals can target larger prey at higher trophic levels, and that their greater diving capabilities [13] also enable them to target more benthic species. These differences may be accentuated by spatial segregation linked to maternal investment, such as females regularly returning to land to feed their pups [8,10,14]. For many seal species, changes in size-related diving capability [15] and experience may explain ontogenetic shifts in foraging ecology [3,9,11,16,17]. The same factors could also explain age-related changes in cetacean diets [18–20]. However, in cetaceans, size differences between males and females are usually smaller, and despite maternal care over several years, many cetacean species show similar foraging ecology between sexes [20–25] (although differences have been observed in sperm whales and killer whales [26–28]).

Here, we investigate sex and size variation in the long-term foraging ecology of the two Arctic toothed cetaceans, belugas (*Delphinapterus leucas*) and narwhals (*Monoceros monoceros*), in Greenland. The two Monodontidae species coexist in West Greenland, while only narwhals are distributed in East Greenland (figure 1*a*). Both species can occur in sex-specific and mixed-sex groups [31–34], and show important maternal investment [35] and sexual size dimorphism. Male belugas and narwhals are 40–100 cm and approximately 60 cm longer than females, respectively [31,36,37], and 40–45% heavier [36,37]. Male narwhals furthermore have an erupted left tooth, the tusk, which is believed to be a secondary sexual trait [38].

The foraging ecology of belugas and narwhals has been studied using stomach contents and stable isotopes of carbon ($\delta^{13}$C) and nitrogen ($\delta^{15}$N) in soft tissues, which have turnover rates of several months [39–41], and are limited to reflect seasonal (often spring/summer) diet [42–44]. $\delta^{13}$C reflects the feeding habitat (pelagic versus benthic, offshore versus coastal) and $\delta^{15}$N the trophic level [45–48]. In Baffin Bay (West Greenland/eastern Canada), Arctic cod (*Boreogadus saida*), polar cod (*Arctogadus glacialis*) and capelin (*Mallotus villosus*) appear to be the main summer prey of both whale species [31,42,43,49]. Interspecific competition between belugas and narwhals is therefore likely in West Greenland, where their distributions overlap [29]. Capelin represent a large proportion of the summer diet of narwhals in East Greenland [43]. Both whale species undertake seasonal migrations, following the distribution of sea ice [29,50]. During winter, they stay in ice-covered areas inaccessible to humans,

limiting dietary information, although Greenland halibut (*Reinhardtius hippoglossoides*), capelin and squid (*Gonatus fabricii*) are believed to be important prey [42,44,49,51]. For both species, sexes show no or little dietary differences (0.2‰) in stable isotope compositions of soft tissues [42,43].

To gain insights into the longer term foraging ecology of beluga and narwhal populations in Greenland, we analysed bone collagen $\delta^{13}$C and $\delta^{15}$N, which reflect an individual's diet over multiple years [52,53]. Studies examining variation in foraging ecology on the basis of isotopic measurements of tissues with rapid turnover rates reflect only a limited portion of the year and, for Arctic species, are consistently biased towards the spring and summer months. By contrast, isotopic analyses of bone collagen offer an opportunity to examine the average diet across seasons. Thus, this approach provides insight into sustained foraging patterns that persist over years, circumventing variation caused by short-term dietary variation. To elucidate differences related to sex and size, we investigated whether beluga and narwhal foraging ecology differs across space.

# 2. Material and methods

## 2.1. Laboratory analyses

We obtained 300 mg of bone powder from the skulls of 27 belugas and 40 narwhals from West Greenland (WG), and 39 narwhals from East Greenland (EG, figure 1*a*, electronic supplementary material, table S1). All samples represented sub-adults or adults. $\delta^{13}$C and $\delta^{15}$N values for 14 WG belugas and 11 WG narwhals were from [54]. The skulls were collected between 1990 and 2007 during subsistence hunting, and are housed at the Natural History Museum of Denmark, University of Copenhagen. Sex information for 100 out of the 106 samples was determined previously by molecular methods [55], an inspection of sexual organs, or the presence/absence of a tusk in narwhals [56]. We followed the laboratory procedures described in [54] and in the electronic supplementary material, text.

## 2.2. Statistical analyses

We ran all statistical analyses in R v. 3.6.1 [57]. Our data satisfied normality and homogeneity of variance for all subdivisions. To test for dietary differences among the three populations (WG belugas, WG narwhals, EG narwhals), we compared their $\delta^{13}$C and $\delta^{15}$N using ANOVA and Tukey's post hoc tests using the package multcomp [58].

To test for dietary differences between males and females, we compared $\delta^{13}$C and $\delta^{15}$N within populations using Student's *t*-tests. The effects of location and sampling date on our results are presented in the electronic supplementary material, text; they had limited impact on our conclusions.

To investigate the association between the size of individuals and their isotopic compositions, we ran a linear regression between skull length (used as a proxy for body length [59]) and $\delta^{13}$C or $\delta^{15}$N within each population. A linear regression was also run separately for males and females (termed groups) within populations. Owing to some of the skulls being fragmented or otherwise broken, we were able to obtain skull measurements for 81 of the 106 belugas and narwhals investigated (see the electronic supplementary material, text for details).

We compared isotopic niche (a proxy for an ecological niche) among populations, and between groups (males/females) within each population, using Bayesian multivariate ellipse-based metrics implemented in the packages SIBER and rjags [60–63]. We calculated standard ellipse areas corrected for sample size (SEA$_C$), and Bayesian standard ellipses (SEA$_B$) for each population/group. We estimated SEA$_B$ using $10^5$ posterior draws, a burning of $10^3$ and a thinning of 10, and used SEA$_B$ to test for differences in niche width among populations/groups (i.e. the proportion (p) of draws of the posterior distribution of the SEA$_B$ in which the area of one population/group was smaller than the other). We evaluated isotopic niche similarity between two populations/groups as the proportion (%) of the non-overlapping area of the maximum-likelihood fitted ellipses of the two.

# 3. Results and discussion

## 3.1. Niche differentiation among populations

Our results indicate resource partitioning of belugas and narwhals in West Greenland. The species occupy distinct, yet slightly overlapping (9%), ecological niches, with belugas showing higher bone

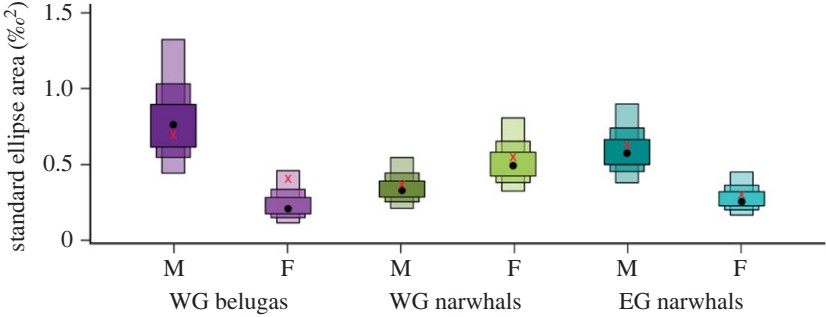

**Figure 2.** Size of the standard ellipse areas (SEA, ‰$^2$) for male and female West Greenland (WG) belugas ($n_M = 14$, $n_F = 9$), WG narwhals ($n_M = 22$, $n_F = 16$) and East Greenland (EG) narwhals ($n_M = 19$, $n_F = 20$), with the black dot indicating the mode of the size of the SEA$_B$, the red cross the mean for the SEA$_C$ and the box edges the 50, 75 and 95% credible intervals.

collagen $\delta^{13}$C and $\delta^{15}$N ($p < 0.01$, electronic supplementary material, figure S3 and table S3). Detailed results are provided in the electronic supplementary material, text. This observed niche partitioning may be the result of interspecific competition, or of differences in dietary preference. Our findings are consistent with studies indicating that—despite prey overlap between species [28,41–43,46]—belugas eat less squid, which are at lower trophic levels [43], than narwhals, and also forage on Atlantic cod (*Gadus morhua*) and other fishes such as redfish (*Sebastes mentella*) [64], which feed at higher trophic levels [65]. Although it is not possible to test for this using our stable isotope data, our findings may reflect that belugas feed on larger individuals, representing higher trophic levels, than narwhals (if both species feed on Greenland halibut or Arctic/polar cod [31,42,43,49,51]).

We did not include prey isotopic data, as these are available from soft tissue samples only [43,44,65] and reflect the short-term seasonal diet of the prey species, while our cetacean stable isotope data from bone collagen reflect the consumer diet over several years. We attempted to incorporate prey data [43,44,65] but the resulting mixing polygon was inconsistent with the isotopic compositions observed in belugas and narwhals after correcting for the Suess effect, which was done for the consumers following Szpak *et al.* 2019 [66] and trophic enrichment [67]. This suggests that the prey isotopic data are not representative of what would have been consumed by these cetaceans either because isotopic compositions are inaccurate, certain prey sources are missing, or a combination of the two. Despite belugas being considered a more generalist feeder [68], we find no significant differences in isotopic niche size between species (electronic supplementary material, figures S3, S4 and table S3).

For narwhals, we find regional differences, with no niche overlap and higher $\delta^{13}$C and $\delta^{15}$N in West Greenland than in East Greenland ($p < 0.01$; electronic supplementary material, table S3 and figure S3; figure 1*b*). This finding supports the long-term habitat segregation of narwhals west and east of Greenland, as has been shown by telemetry studies [50], and is also in agreement with stable isotope analysis of skin tissue; Watt *et al.* 2013 showed that Baffin Bay narwhals, whose main prey are benthic fish species including halibut, Arctic cod and polar cod, have a less pelagic diet than East Greenland narwhals, which consume a larger proportion of capelin [43]. Higher $\delta^{13}$C and $\delta^{15}$N in West relative to East Greenland is also reported in fishes [43] and other marine mammal species [69], suggesting regional differences at the base of the food web.

## 3.2. Sex and size differences in foraging ecology

While $\delta^{13}$C does not differ significantly between sexes in any population ($p > 0.05$), our results indicate sex and size differences in the foraging ecology of West Greenland belugas and East Greenland narwhals based on $\delta^{15}$N (figure 1*b*; electronic supplementary material, figure S5). Males have significantly higher $\delta^{15}$N (WG belugas $p = 0.03$, EG narwhals $p = 0.01$) and larger ecological niches than females ($p \geq 0.99$; electronic supplementary material, table S3; figures 1*b* and 2). This contrasts with the analysis of $\delta^{13}$C and $\delta^{15}$N obtained from skin samples, which did not show any difference in niche size between sexes in narwhals in both Baffin Bay and East Greenland [43]. These discrepancies between stable isotope values estimated from bone and skin probably reflect the more rapid turnover rate of skin (months versus years), and therefore short-term versus long-term ecological differences between sexes owing to their differing energetic and physiological needs (see below). Moreover, bone collagen stable isotope data reflect year-round foraging, averaged over several years, including the winter, where most of the

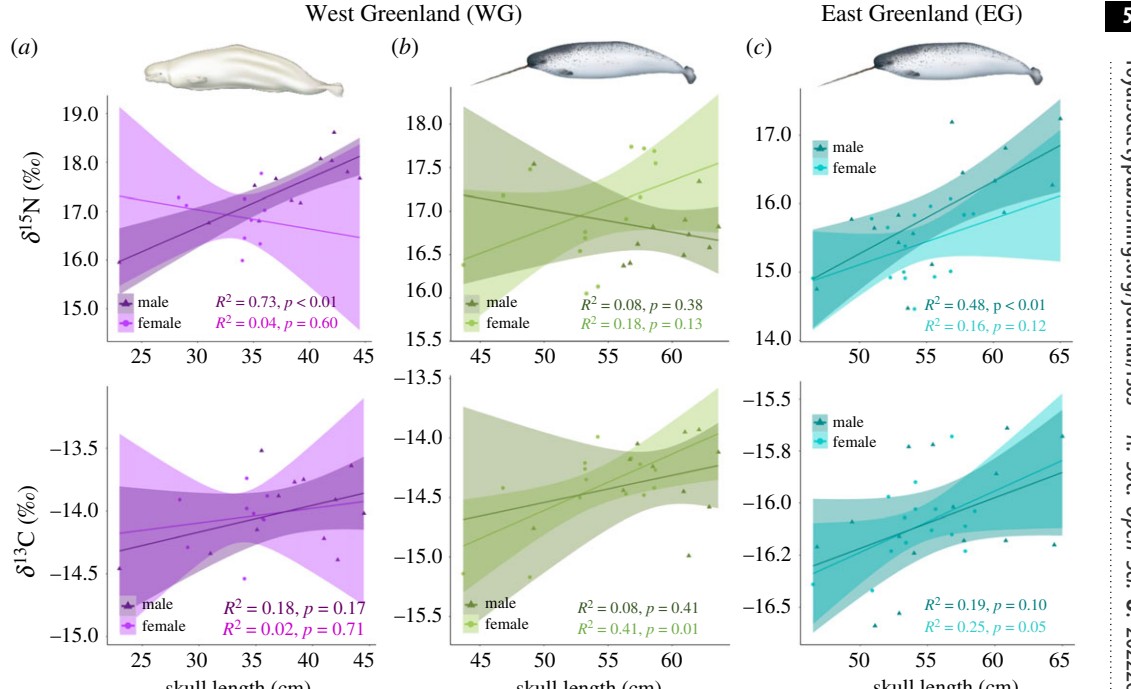

**Figure 3.** Variation in $\delta^{13}$C and $\delta^{15}$N from bone samples according to skull length (cm) for male and female (*a*) WG belugas ($n_M = 12$, $n_F = 9$), (*b*) WG narwhals ($n_M = 11$, $n_F = 14$) and (*c*) EG narwhal ($n_M = 15$, $n_F = 16$). The regression lines for each sex and their confidence intervals (shading) are plotted, with females in a lighter colour shade than males. Beluga and narwhal illustrations are from Uko Gorter.

foraging occurs for narwhals [49], and which is not reflected in stable isotope data from skin tissue collected in the summer.

Larger belugas and East Greenland narwhals tend to have higher $\delta^{13}$C ($p = 0.05$ and $p = 0.01$, respectively) and higher $\delta^{15}$N ($p < 0.01$; electronic supplementary material, figure S5). The pattern is stronger than that reported using skin tissue from belugas sampled in southeast Baffin Island, where older individuals show marginally higher ($<0.5‰$) $\delta^{13}$C and $\delta^{15}$N [42], again possibly owing to differences in integration times between skin and bone collagen.

We hypothesize the sex-specific differences in foraging ecology observed in West Greenland belugas and East Greenland narwhals are driven by a combination of sexual size dimorphism and maternal investment, as has been observed in pinnipeds [3,8,9,16]. Larger individuals may be able to dive deeper, as they have a greater capacity to store oxygen in their tissues [15,70], allowing them to target higher trophic-level benthic prey. Larger belugas spend significantly less time at the surface than smaller individuals [71]. Their larger size may also enable them to catch and handle larger individuals of the same prey species that are at higher trophic levels (e.g. halibut or cod). Alternatively, fine-scale habitat segregation between sexes, as observed in Beaufort Sea belugas, could explain sex differences in diet [72].

The significantly positive correlation between trophic level and size is observed in male belugas and male East Greenland narwhals ($p < 0.01$; figure 3). The absence of a similar pattern in females may reflect their maternal investment. Despite female odontocetes raising their calves for several years and investing significant energy resources in their growth and survival, many species show no apparent sex-based dietary differences [20–25]. The offshore and deep-diving lifestyle of both belugas and narwhals may exacerbate sex-specific differences in feeding ecology, as females, whatever their size, may (at least for periods of time) adjust their diving and foraging behaviour to that of their calves, which have lower aerobic capabilities [73]. Both belugas and narwhals are predated by killer whales [74], and females probably invest in protecting their calves from this predator. Our hypothesis correlates well with studies of diving behaviour. In eastern Canada, female belugas with calves spend a higher proportion of time at the surface than females without calves and have lower diving rates [71].

Maternal investment may explain the more pronounced sex differences in foraging ecology obtained from stable isotope analysis of bone collagen than of skin tissue [42,43]. All females may not be accompanied by a calf during the particular season (often spring/summer) represented by a soft

tissue sample, whereas most adult females probably have had a calf (or multiple calves) over the integration of time represented by bone collagen.

## 3.3. The lack of sex and size variation in diet in West Greenland narwhals

We find no evidence of differentiated foraging ecology between males and females in narwhals in West Greenland ($p > 0.05$), and the size of the ecological niche is similar between sexes (p = 0.11; figures 1*b* and 2). Furthermore, we find a correlation between length and $\delta^{13}$C ($p = 0.01$, electronic supplementary material, figure S5) only, in particular in females (figure 3), with no size differences in $\delta^{15}$N ($p = 0.79$; figure 3; electronic supplementary material, figure S5), suggesting sex and size variation in diet may be population specific.

The pattern may reflect interspecific competition with belugas in areas where the species distributions overlap. The niche of a species and within-population variation in ecology may be constrained by interspecific competition [2]. Belugas may potentially target larger prey at higher trophic levels, and hence being a larger male narwhal would present no advantage for accessing higher trophic-level resources. Alternatively, a lack of differentiation in stable isotope values does not necessarily imply there are no dietary differences, as different prey can have similar isotopic values. Therefore, there could be sex and size differences in diet, linked to the same factors as for the other populations, including maternal investment, but which we do not detect with our data.

We investigated whether combining sampling across four locations (Qaanaq, Melville Bay, Uummannaq and Qeqertarsuaq, figure 1*a*) for West Greenland narwhals influenced our results (electronic supplementary material, text). We do not observe any differences between sexes within the one locality (Qaanaaq) from which we had a sufficient sample size for sex comparison (see the electronic supplementary material, text). Thus, including several localities probably does not influence our findings of a lack of sex differences in the foraging ecology of West Greenland narwhals. Among certain localities, we recover variation in $\delta^{15}$N, although we observe no variation in $\delta^{13}$C (electronic supplementary material, figure S6). This may reflect geographical partitioning within the large West Greenland (i.e. Baffin Bay) population, which is composed of several stocks [29]; every year, different stocks of narwhals in West Greenland migrate between various coastal summer localities, and offshore in Baffin Bay during winter [50]. We suggest such geographical resource partitioning may be another mechanism to decrease intraspecific competition.

## 4. Conclusion

We report long-term differences in foraging ecology between sexes in West Greenland belugas and East Greenland narwhals. These findings probably reflect a combination of sexual size, dimorphism, maternal investment and a deep-diving lifestyle. However, we find no differences in diet between sexes in West Greenland narwhals, indicating that sex differences in foraging ecology are population specific, and may be driven by intra- and interspecific competition. Our study highlights the applicability of stable isotope analysis of bone collagen for revealing long-term dietary differences among individuals with different life histories or physiological needs, which may remain undetected in the analysis of soft tissues with shorter turnover.

Data accessibility. Electronic supplementary material, table S1.

Authors' contributions. E.D.L. conceived the study. M.L., P.S. and E.D.L. conceived the analyses. E.G. and M.P.H.-J. collected samples. M.L. and M.S. sampled the skulls. P.S. ran the isotopic laboratory analyses. M.L. performed the statistics. M.L. and E.D.L. wrote the manuscript with input from all co-authors.

Competing interests. We have no competing interests.

Funding. This research was supported by the Carlsberg Foundation Distinguished Associate Professor Fellowship, grant CF16-0202 to E.D.L.

Acknowledgements. We thank fieldworkers and hunters involved in specimen collection. We are grateful to Uko Gorter for the illustrations, Julie Lorenzen for sampling advice and Rene Swift for figure 1*a*. We also would like to thank three anonymous reviewers for their input.

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
