## [Peer Review File · Royal Society Open Science]

Review History

RSOS-202226.R0 (Original submission)

Review form: Reviewer 1

Is the manuscript scientifically sound in its present form?

Yes

Are the interpretations and conclusions justified by the results?

Yes

Is the language acceptable?

Yes

Do you have any ethical concerns with this paper?

No

Have you any concerns about statistical analyses in this paper?

No

Recommendation?

Accept with minor revision (please list in comments)

Comments to the Author(s)

I have re-reviewed (I was reviewer 2 in Biology Letters submission) the paper by Louis et al. on sex- and size-based variation in foraging ecology of belugas and narwhals across space. They have done a great job incorporating previous reviewer comments and I only have one minor comment. I look forward to seeing this article published in Royal Society Open Science.

Line 87-90 – In this objectives sentence - I suggesting removing the inference in the ability to test for differences in foraging ecology of beluga and narwhal in relation to interspecific competition. Again, this is not directly tested in this study due to the lack of prey availability and prey abundance data. I think it's fine to have the listed objective as, "To elucidate differences related to sex and size, we investigated whether beluga and narwhal foraging ecology differs across space" and to just speculate on the potential competition aspects between beluga and narwhal in the discussion.

Review form: Reviewer 2

Is the manuscript scientifically sound in its present form?

Yes

Are the interpretations and conclusions justified by the results?

Yes

Is the language acceptable?

Yes

Do you have any ethical concerns with this paper?

No

Have you any concerns about statistical analyses in this paper?

No

Recommendation?

Accept as is

Comments to the Author(s)

After reading the revised manuscript by Louis et al., I maintain my earlier assessment, that the paper is concise, well-written and neatly presented. I found the methods and statistical analyses to be appropriate and well-conducted.

I appreciate the thoughtful and thorough manner with which the authors responded to my previous general and specific comments. They have adequately addressed the concerns I had and therefore I have no comments on the revised version.

I believe that the manuscript represents a valuable contribution to the literature on cetacean resource use.

Decision letter (RSOS-202226.R0)

Dear Dr Louis

On behalf of the Editors, we are pleased to inform you that your Manuscript RSOS-202226 "Population-specific sex and size variation in long-term foraging ecology of belugas and narwhals" has been accepted for publication in Royal Society Open Science subject to minor revision in accordance with the referees' reports. Please find the referees' comments along with any feedback from the Editors below my signature.

Please submit your revised manuscript and required files (see below) no later than 7 days from today's (ie 06-Jan-2021) date. Note: the ScholarOne system will 'lock' if submission of the revision is attempted 7 or more days after the deadline. If you do not think you will be able to meet this deadline please contact the editorial office immediately.

on behalf of Prof Pete Smith (Subject Editor)
openscience@royalsociety.org

Reviewer comments to Author:
Reviewer: 1

Comments to the Author(s)

I have re-reviewed (I was reviewer 2 in Biology Letters submission) the paper by Louis et al. on sex- and size-based variation in foraging ecology of belugas and narwhals across space. They have done a great job incorporating previous reviewer comments and I only have one minor comment. I look forward to seeing this article published in Royal Society Open Science.

Line 87-90 – In this objectives sentence - I suggesting removing the inference in the ability to test for differences in foraging ecology of beluga and narwhal in relation to interspecific competition. Again, this is not directly tested in this study due to the lack of prey availability and prey abundance data. I think it's fine to have the listed objective as, "To elucidate differences related to sex and size, we investigated whether beluga and narwhal foraging ecology differs across space" and to just speculate on the potential competition aspects between beluga and narwhal in the discussion.

Reviewer: 2

Comments to the Author(s)

After reading the revised manuscript by Louis et al., I maintain my earlier assessment, that the paper is concise, well-written and neatly presented. I found the methods and statistical analyses to be appropriate and well-conducted.

I appreciate the thoughtful and thorough manner with which the authors responded to my previous general and specific comments. They have adequately addressed the concerns I had and therefore I have no comments on the revised version.

I believe that the manuscript represents a valuable contribution to the literature on cetacean resource use.

===PREPARING YOUR MANUSCRIPT===

===PREPARING YOUR REVISION IN SCHOLARONE===

Author's Response to Decision Letter for (RSOS-202226.R0)

See Appendix A.

Decision letter (RSOS-202226.R1)

Dear Dr Louis,

It is a pleasure to accept your manuscript entitled "Population-specific sex and size variation in long-term foraging ecology of belugas and narwhals" in its current form for publication in Royal Society Open Science.

on behalf of Prof Pete Smith (Subject Editor)
openscience@royalsociety.org

Follow Royal Society Publishing on Twitter: @RSocPublishing
Follow Royal Society Publishing on Facebook:
<https://www.facebook.com/RoyalSocietyPublishing.FanPage/>

Read Royal Society Publishing's blog:
<https://royalsociety.org/blog/blogsearchpage/?category=Publishing>

Appendix A

Dear Dr Louis

On behalf of the Editors, we are pleased to inform you that your Manuscript RSOS-202226 "Population-specific sex and size variation in long-term foraging ecology of belugas and narwhals" has been accepted for publication in Royal Society Open Science subject to minor revision in accordance with the referees' reports. Please find the referees' comments along with any feedback from the Editors below my signature.

Please submit your revised manuscript and required files (see below) no later than 7 days from today's (ie 06-Jan-2021) date. Note: the ScholarOne system will 'lock' if submission of the revision is attempted 7 or more days after the deadline. If you do not think you will be able to meet this deadline please contact the editorial office immediately.

on behalf of Prof Pete Smith (Subject Editor)
openscience@royalsociety.org

>Dear Editor,

We are pleased to hear that our manuscript is accepted. Please find below our responses to the reviewers' comments.

On behalf of the co-authors,

Marie Louis

Reviewer comments to Author:

Reviewer: 1

Comments to the Author(s)

I have re-reviewed (I was reviewer 2 in Biology Letters submission) the paper by Louis et al. on sex- and size-based variation in foraging ecology of belugas and narwhals across space. They have done a great job incorporating previous reviewer comments and I only have one minor comment. I look forward to seeing this article published in Royal Society Open Science.

Line 87-90 – In this objectives sentence - I suggesting removing the inference in the ability to test for differences in foraging ecology of beluga and narwhal in relation to interspecific competition. Again, this is not directly tested in this study due to the lack of prey availability and prey abundance data. I think it's fine to have the listed objective as, "To elucidate differences related to sex and size, we investigated whether beluga and narwhal foraging ecology differs across space" and to just speculate on the potential competition aspects between beluga and narwhal in the discussion.

>Dear reviewer,

Thank you for your positive feedback.

We have made the change in the objective sentence L88-90 and it now reads as you suggest: "To elucidate differences related to sex and size, we investigated whether beluga and narwhal foraging ecology differs across space."

On behalf of the co-authors,

Marie Louis

Reviewer: 2

Comments to the Author(s)

After reading the revised manuscript by Louis et al., I maintain my earlier assessment, that the paper is concise, well-written and neatly presented. I found the methods and statistical analyses to be appropriate and well-conducted.

I appreciate the thoughtful and thorough manner with which the authors responded to my previous general and specific comments. They have adequately addressed the concerns I had and therefore I have no comments on the revised version.

I believe that the manuscript represents a valuable contribution to the literature on cetacean resource use.

>Dear reviewer,

Thank you for your positive feedback.

On behalf of the co-authors,

Marie Louis